# Anti-Cholinesterase and Anti-α-Amylase Activities and Neuroprotective Effects of Carvacrol and *p*-Cymene and Their Effects on Hydrogen Peroxide Induced Stress in SH-SY5Y Cells

**DOI:** 10.3390/ijms24076073

**Published:** 2023-03-23

**Authors:** Lucia Caputo, Giuseppe Amato, Laura De Martino, Vincenzo De Feo, Filomena Nazzaro

**Affiliations:** 1Department of Pharmacy, University of Salerno, Via Giovanni Paolo II, 132, 84084 Fisciano, Italy; 2Institute of Food Sciences, CNR-ISA, Via Roma, 64, 83100 Avellino, Italy

**Keywords:** carvacrol, *p*-cymene, Alzheimer’s disease, diabetes mellitus, natural compounds

## Abstract

Several researchers have demonstrated the health and pharmacological properties of carvacrol and *p*-cymene, monoterpenes of aromatic plants. This study investigated these compounds’ possible anti-cholinesterase, anti-α-amylase, and neuroprotective effects. We evaluated the anti-acetylcholinesterase and anti-α-amylase activities at different concentrations of the compounds. The maximum non-toxic dose of carvacrol and *p*-cymene against SH-SY5Y neuroblastoma cells was determined using an MTT assay. The neuroprotective effects of the compounds were evaluated on H_2_O_2_-induced stress in SH-SY5Y cells, studying the expression of caspase-3 using Western blotting assays. Carvacrol showed inhibitory activities against acetylcholinesterase (IC50 = 3.8 µg/mL) and butyrylcholinesterase (IC_50_ = 32.7 µg/mL). Instead, the anti-α-amylase activity of carvacrol resulted in an IC_50_ value of 171.2 μg/mL After a pre-treatment with the maximum non-toxic dose of carvacrol and *p*-cymene, the expression of caspase-3 was reduced compared to cells treated with H_2_O_2_ alone. Carvacrol and *p*-cymene showed in vitro anti-enzymatic properties, and may act as neuroprotective agents against oxidative stress. Further studies are necessary to elucidate their possible use as coadjutants in preventing and treating AD in diabetic patients.

## 1. Introduction

Medicinal plants can be essential in treating several disorders, including Alzheimer’s disease (AD) and diabetes mellitus [1]. AD is an age-related progressive neurodegenerative disease of the central nervous system, and it is considered as one of the most common forms of dementia. The main neuropathological hallmark of AD is an abnormal accumulation of extracellular β-amyloid (Aβ) protein that forms neuritic plaques [2], mainly composed of the microtubule-associated protein Tau in a hyperphosphorylated form [3]. In addition, an increased acetylcholinesterase (AChE) expression, metabolic disorders, and oxidative stress are linked to the etiology of AD [4]. AChE is a key enzyme in the cholinergic nervous system. AChE may interact with β-amyloid peptide and increase its aggregation and deposition into insoluble plaques [5]. However, the possible role of AChE in the development of a vicious Aβ cycle and P-tau dysregulation has also been reported [6]. Although the inhibition of AChE has been extensively studied as a symptomatic treatment in neurological diseases, less attention has been paid to its sister enzyme, butyrylcholinesterase (BChE), that co-regulates the metabolism of the neurotransmitter acetylcholine [7]. The main difference between the two enzymes is their localization: AChE is localized mainly into neurons, and BChE is associated primarily with glial cells, endothelial cells, and neurons [8]. High BChE levels can be associated with neuropathologic hallmarks of AD, such as neuritic plaques and neurofibrillary tangles [9]. Therefore, discovering selective BChE inhibitors warrants drug development for AD treatment [10]. Several drugs are recommended to enhance memory, such as the leaves of *Ginkgo biloba* L. (Ginkoaceae) and the fruits of *Lycium barbarum* L. (Solanaceae) [11], or to treat dementia-related disorders. For example, Ozarowski and coworkers [12] reported that an extract from leaves of *Rosmarinus officinalis* L. (Labiatae) improved memory in rat brain by AChE inhibition.

Moreover, recent studies suggested a relationship between AD and type II diabetes mellitus (T2DM), a metabolic disorder characterized by chronic hyperglycemia and proposed as an independent risk factor for AD [13]. The two diseases share similar pathophysiological pathways: both are characterized by amyloid deposition [14], neurodegeneration [15], and an increase in AChE activity [16]. Moreover, α-amylase, a key enzyme that hydrolyzes starch molecules to give polymers composed of glucose units and causes hyperglycemia and the development of T2DM [17], is also expressed and active in the human brain: AD patients show an increased activity of this enzyme compared to non-AD patients [18]. Medicinal plants are commonly considered as alternative therapies for treating diabetes mellitus: some of them are effective in controlling the plasmatic level of glucose with minimal side effects [19]. AChE inhibition can be a strategy used to treat AD, but the few existing synthetic drugs for treating cognitive disorders have many side effects [20]. Natural plant-derived substances can represent interesting alternatives to synthetic molecules; in particular, essential oils (EOs) and their main components have great potential for the treatment of AD [21] and diabetes mellitus [22]. Previously, our research group found that linalool, one of the main components of the essential oils from some aromatic plants, inhibited the expression of proteins such as pERK and PKA in human neuroblastoma SH-SY5Y cancer cells [23] and were protective against amyloid β-neurotoxicity [24].

Plants with high levels of carvacrol and/or *p*-cymene have been used in traditional medicine in many parts of the world due to the activities shown by these two compounds [25].

Carvacrol is a monoterpene, isomer of thymol, present in essential oils derived from an aerial part of some aromatic plants belonging, among others, to *Corydothymus, Origanum, Satureja*, and *Thymus* genera [26]. This compound shows strong antimicrobial activity against both Gram-positive and Gram-negative bacteria [27], besides antioxidant, neuroprotective, and antiviral properties [28]. 

*p*-Cymene is a monocyclic monoterpene commonly found in essential oils of several aromatic plants species from the genera *Artemisia, Origanum, Ocimum, Thymus*, and *Eucalyptus* [25]. The compound has been reported for several pharmacological activities, such as antimicrobial, antiviral, antioxidant, anti-inflammatory, and antidiabetic [29,30,31]. Moreover, De Oliveira and coworkers showed that *p*-cymene may act as a neuroprotective agent in the brain [32].

In light of previous studies that highlight standard features between AD and T2DM and consider medicinal plants or their constituents as potential neuroprotective agents [21,33] and enzymatic inhibitors [17,21], and considering that, to the best of our knowledge, there are no studies on natural substances that correlate neuroprotective effects with possible activity against enzymes involved in diabetes, the aims of this study were: (1) to evaluate the possible effects of carvacrol and its precursor *p*-cymene on AChE, BChE, and α-amylase activities; (2) to study the potential of these monoterpenes as neuroprotective agents in hydrogen-peroxide-induced stress in SH SY5Y cells, considering that, in neurodegenerative diseases, oxidative brain damage is often present.

## 2. Results and Discussion

### 2.1. Cholinesterase Inhibitory Activity

The possible inhibitory effects of carvacrol and *p*-cymene on cholinesterases (AChE and BChE) activity were evaluated in this study (Table 1). The AChE inhibitory effect exerted by carvacrol was four times stronger than that exerted by *p*-cymene, with an IC_50_ value of 3.8 μg/mL, although carvacrol and *p*-cymene have a very similar chemical structure. Carvacrol was also more active than *p*-cymene in inhibiting BChE, with an IC_50_ = 32.7 μg/mL.

### 2.2. α-Amylase Inhibitory Activity

The inhibition of α-amylase activity is a strategy used to lower postprandial blood glucose levels in the case of T2DM [34]. However, it is appropriate to study new possible therapeutic approaches to avoid the severe side effects of anti-diabetic drugs used. Our results show IC_50_ values of 171.1 and 215.2 μg/mL for carvacrol and *p*-cymene, respectively.

### 2.3. Hydrogen Peroxide Scavenging Activity

The H_2_O_2_ scavenging activity of carvacrol and *p*-cymene was evaluated to investigate their possible neuroprotective effects on hydrogen-peroxide-induced stress in SH-SY5Y cells. The results are reported in Figure 1. Carvacrol was more active than *p*-cymene: at the maximum concentration tested (1000 μg/mL), this compound allowed the primary formation of the Fe2+-tri-phenanthroline complex compared with *p*-cymene.

### 2.4. Determination of the Maximum Non-Toxic Dose (MNDT)

The MTT assay determined the MNDT of carvacrol and *p*-cymene on SH-SY5Y cells (Figure 2). The compounds did not affect the cell viability at a concentration ≤ 50 μg/mL.

### 2.5. Determination of the Optimal Concentration of Hydrogen Peroxide and Caspase 3 Expression

The ability to inhibit cell growth and the treatment time were considered as two critical parameters in determining the optimal concentration of H_2_O_2_ to be used in the test. Hydrogen peroxide is an oxidative stress inducer in several cell lines; the optimal concentration and treatment time used varied in each study according to the objectives [35]. Our study selected the optimal concentration of H_2_O_2_ and the time treatment to obtain a cell viability of at least 80%. The cell viability was determined using the MTT test after two and four h of treatment (Figure 3). After these preliminary experiments, 100 and 200 μM represented the concentrations chosen, and two h was the appropriate treatment time. The SH-SY5Y cells were pretreated with carvacrol and *p*-cymene to understand if these substances could exert a neuroprotective effect on H_2_O_2_-induced stress; subsequently, the cells were treated with H_2_O_2_ for 2 h; then, we performed their lysis and protein extraction. The results demonstrate that caspase-3 expression significantly increased in neuroblastoma cells treated with 100 and 200 mM H_2_O_2_ for 2 h compared to control cells (Figure 4 and Figure 5). Furthermore, following pretreatment with carvacrol (50 μg/mL) for 22 h, caspase-3 was inhibited, and its expression was similar to that of untreated cells and cells treated with carvacrol alone (50 μg/mL) (Figure 4). In contrast, in the pretreatment with *p*-cymene (50 μg/mL), the expression of caspase-3 increased similarly to the cells treated with 100 or 200 mM H_2_O_2_. Our results demonstrate that carvacrol exerted a more significant neuroprotective effect on H_2_O_2_-induced stress than *p*-cymene, confirming the results of the previous cell-free assay on H_2_O_2_ scavenging activity.

Our results are well inserted in this scenario. In fact, beyond a good antioxidant activity, carvacrol and *p*-cymene also demonstrated anti-acetylcholinesterase and anti-α-amylase activities with respect to positive controls used in assays.

## 3. Discussion

### 3.1. Cholinesterase Inhibitory Activity

Cholinesterase and α-amylase synthetic inhibitors are often used as AD treatments. However, they have several limitations due to a short half-life and too many side effects, such as hepatic and gastrointestinal disturbances [17]. Carvacrol and *p*-cymene have often been investigated for their potential insect-derived AChE inhibitory activity [34]. The results show that carvacrol was more active than *p*-cymene in inhibiting AChE and BChE, with an IC50 = 32.7 μg/mL. Moreover, literature data reported that carvacrol was 10 times stronger than that exerted by its isomer thymol, although thymol and carvacrol have a very similar structure; in fact they only differ in terms of their OH-group position [35]. Thymol also inhibited BChE in a dose-dependent manner but, at a concentration of 2 mg/mL, only inhibited 42.2% of BChE activity [36].

Furthermore, carvacrol shows AChE inhibitory potential [37], which helps to treat neurological disorders such as AD. Our results agreed with data reported in the literature on the possible activity of carvacrol against BChE [38]. Unfortunately, no data are available in the literature about the anti-AChE and BChE activities of *p*-cymene. Recently, the anti-acetylcholinesterase activity of some essential oils, with carvacrol and *p*-cymene as the main constituents, was reported [39]. Terpenoids, which are among the main components of essential oils, are small lipophilic molecules that, when inhaled, can be absorbed through the nasal mucosa or, when applied to the skin, can enter the blood and cross the blood–brain barrier [40]. Zotti and collaborators [41] suggested that carvacrol was an active compound in the brain capable of influencing neuronal activity through the modulation of neurotransmitters: its mechanism of action remains unclear.

### 3.2. α-Amylase Inhibitory Activity

The results show that carvacrol and *p*-cymene had a similar activity against α-amylase, with an IC50 of 171.1 and 215.2 μg/mL, respectively. Recently, Siahbalaei and coworkers [42] reported that some essential oils, rich in carvacrol, displayed vigorous antioxidant activity against glucose oxidation and anti-diabetic effects against amylase and glucosidase activities, suggesting their possible anti-diabetic potential. By molecular docking analysis, Pathak and coworkers [43] demonstrated that carvacrol had a good docking score toward the α-amylase enzyme. In an l-arginine-induced pancreatitis animal model, a dose of 10 mg/kg of carvacrol was able to prevent an α-amylase increase [44]; in addition, in rats with acute pancreatitis, the lipase and amylase levels were reduced in the group treated with carvacrol with respect to the control [45]. Interestingly, a phase I study regarding carvacrol showed clinical safety and tolerability in healthy subjects [46]. Moreover, potential antidiabetic activity has been reported for thymol extracted from several plant species: *Thymus quinquecostatus*, *T. linearis*, and *T. serrulatus* [47,48,49]. No previous studies have reported the possible activity of *p*-cymene.

### 3.3. Hydrogen Peroxide Scavenging Activity

The antioxidant effect of carvacrol was widely reported in both in vitro and in vivo studies [29]; it can improve the activity of enzymatic and non-enzymatic antioxidants [50], and could be an alternative to synthetic antioxidants [51]. In addition, phenols, such as carvacrol, are well-known antibacterial compounds against Gram-negative and Gram-positive strains [29]. The phenolic hydroxylic group of carvacrol seems essential for antimicrobial activity, probably destabilizing the membrane and depleting the microbial pools of ATP, impairing essential processes and ultimately leading to cell death [52]. In addition, carvacrol has also shown antioxidant and antiviral properties [28]. *p*-Cymene has excellent antioxidant potential in vivo and may act as a neuroprotective agent in the brain [32]. Moreover, this compound increased neurogenesis and reduced amyloid plaque counts in AD rats thanks to its antioxidant and anti-inflammatory properties [53]. No previous studies have highlighted the possible scavenging activity of these two compounds against H_2_O_2_ in vitro or in vivo.

### 3.4. Determination of the Maximum Non-Toxic Dose (MNDT)

Many studies have shown that carvacrol inhibited the proliferation of several cell lines, such as the human hepatocellular carcinoma cell line (HepG-2), non-small-cell lung cancer cells (A549), human breast cancer cell line (MCF-7), human metastatic breast cancer cell line (MDA-MB-231), human colon cancer cell lines (HCT116 and LoVo), and human gastric adenocarcinoma (AGS) [54]. *p*-Cymene also inhibited the proliferation of several tumor cells lines, such as MCF-7, breast carcinoma (MDA-MB-453), colon carcinoma (SW-480), myeloma multiple cells (IM9) [55], gastric carcinoma (SGC-7901), liver carcinoma (BEL-7404), and nasopharyngeal carcinoma (CNE-1) [56]. However, no data are available on SH-SY5Y cells. The Expert Panel of the Flavour and Extract Manufacturers Association (FEMA) reported the generally recognized safe (GRAS) status of carvacrol [57]. Clinical studies on the bioavailability, active forms, and target tissues of carvacrol are needed: in fact, its bioavailability largely depends on the animal model used [29]. As for *p*-cymene, one of its main limitations for pharmaceutical applications is its short half-life [58]: the compound is rapidly absorbed into the circulation system and quickly eliminated in vivo [59]. In our previous study, the cytotoxicity of *Origanum vulgare* essential oil rich in thymol (76%) was evaluated, and the results highlighted a higher toxicity on SH-SY5Y cells (50.5 µg/mL) with respect to carvacrol, probably due to the presence of thymol [60].

To date, few studies have reported the neuroprotective effect of carvacrol. A recent study showed that carvacrol can increase the cell viability of differentiated SH-SY5Y cells and exhibited a protective effect against oxidative stress by preventing Aβ1–42-induced cytotoxicity, LDH release, and increments in malondialdehyde and H_2_O_2_ levels in vitro [61]. Other literature data on the neuroprotective activity of carvacrol suggested that this compound can protect neuroblastoma SH-SY5Y cells against Fe^2+^-induced apoptosis [62] and reduces cadmium-triggered oxidative stress in PC12 cells and caspase-dependent and independent apoptosis [63]. Carvacrol also promotes marked neuroprotection in the hemiparkinsonian mouse model, reducing caspase to basal levels [64]. Chenet and coworkers (2019) demonstrated that a pretreatment of 4 h with carvacrol promotes mitochondrial protection in the human neuroblastoma cells SH-SY5Y exposed to hydrogen peroxide by a mechanism involving heme oxygenase-1 [65]. A previous study suggested that *p*-cymene reduced the formation of reactive oxygen species: in mice, the treatment with *p*-cymene significantly reduced the level of lipid peroxidation [32]. Thymol showed a reduction in cytotoxicity induced by H_2_O_2_ in cortical neurons and in PC12 cells [36,66].

These findings appear of interest considering that, in cases where the cellular metabolic activity is greater than the antioxidant one or in cases where there is an increase in the production or accumulation of free radicals, the oxidative stress, together with the accumulation of proteins and the change in insulin action, is one of the factors linking T2DM and AD [67]. Several studies have shown a possible correlation between DM and dementia, diseases that can share common cellular and molecular mechanisms [67]. Moreover, some diabetic patients are more susceptible to AD than healthy subjects [68]. The brain is an insulin-sensitive organ, where insulin has a neuroprotective function [67], supports neuronal plasticity and cholinergic functions, and plays a fundamental role in learning and memory [69]. Consequently, damaged insulin signaling in the brain can significantly affect cognitive impairment and neurodegeneration [69]. For these strong relationships, recently, some authors have designated Alzheimer’s disease as “diabetes of the brain” or “type 3 diabetes (T3D)” [70], which corresponds to a chronic insulin resistance related to an insulin deficiency limited primarily to the brain [71]. The effectiveness of carvacrol could be attributed to a hydroxyl functional group and a cloud of delocalized electrons. Moreover, the weaker activities of *p*-cymene seem to confirm the importance of the OH group in the phenolic ring. Andrade-Ochoa and coworkers [72] studied the quantitative structure–activity relationship of some essential oils’ constituents, between which, for the biological activity of carvacrol and *p*-cymene, the key function of the hydroxyl group present in carvacrol, its position in the aromatic ring, its relative position to the larger aliphatic chain, and the conformation of the aromatic ring were highlighted.

## 4. Materials and Methods

### 4.1. Reagents

Carvacrol (≥97%, GC); *p*-cymene (≥97%, GC); acetylcholinesterase (AChE) type VI-S from *Electrophorus electricus*, EC 3.1.1.7, 245 U/mg solid; butyrylcholinesterase (BChE) from equine serum 8.8 U/mg; DTNB [5,5′-dithiobis-(2-nitrobenzoic acid)]; acetylthiocholine iodide (AChI); butyrylthiocholine iodide (BChI); galantamine (USP, reference standard); starch azure; porcine pancreatic amylase; calcium chloride; acetic acid; gallic acid, hydrogen peroxide (1 M); hydrocloridric acid; acetone; *N*,*N*-dimethylformamide, 1,10-phenanthroline; ferrous ammonium sulfate; RPMI; l-glutamine; FBS; penicillin/streptomycin; MTT; SDS; primary anti-caspase-3 antibodies were purchased from Sigma-Aldrich (St. Louis, MO, USA). Anti-GAPDH antibody and enhanced chemiluminescence reagents were purchased from Santa Cruz Biotechnology (Santa Cruz, CA, USA); horseradish peroxidase-conjugated secondary antibody was purchased from Amersham Biosciences (Pittsburgh, PA, USA). Nitrocellulose was purchased from Biorad (Hercules, CA, USA).

### 4.2. Cholinesterase Inhibition

The cholinesterase inhibition was evaluated by Ellman’s colorimetric method [73] with some modifications. Briefly, in a total volume of 1 mL, 415 µL of Tris-HCl buffer 0.1 M (pH 8), 10 µL of a buffer solution of carvacrol or *p*-cymene (in methanol) at different concentrations (100, 10, 1 and 0.1 µg/mL), and 25 µL of a solution containing 0.28 U/mL of AChE (or BChE) were incubated for 15 min at 37 °C. Then, a solution of AChI (or BChI) 1.83 mM (75 µL) and 475 µL of DTNB was added, and the final mixture was incubated for 30 min at 37 °C. The absorbance was measured at 405 nm in a spectrophotometer (Thermo Scientific Multiskan GO, Monza, Italy). Galantamine was the positive control.

### 4.3. α-Amylase Inhibition

The α-amylase inhibition was evaluated according to the method of Dineshkumar and coworkers [74]. Briefly, starch azure (10%) was suspended in 0.5 M Tris-HCl buffer (pH 6.9) containing 0.01 M calcium chloride (substrate), boiled for 5 min, and then preincubated at 37 °C for 5 min. Next, carvacrol or *p*-cymene was dissolved in 0.1% dimethyl sulfoxide to obtain final concentrations of 500, 100, 50, 10, and 1 μg/mL. Then, 0.2 mL of carvacrol or *p*-cymene solution was added to the substrate solution tube. Next, 0.1 mL of α-amylase (2 U/mL) in Tris-HCl buffer was added to the tube containing carvacrol or *p*-cymene solution and substrate solution. The reaction was carried out at 37 °C for 10 min. Then, 500 µL of acetic acid solution (50%) was added, and the mixture was centrifuged at 2000 rpm for 5 min. The absorbance of the supernatant was measured at 595 nm using a spectrophotometer (Thermo Scientific Multiskan GO, Monza, Italy). Acarbose was used as a positive control.

### 4.4. Enzyme Activity

The percent inhibition of enzyme activity for acetylcholinesterase, butyrylcholinesterase, and α-amylase was calculated by comparison with the absorbance of the control without sample, following the formula: % = [(A_0_ − A_1_)/A_0_] × 100(1)
where A_0_ is the absorbance of the control without the sample and A_1_ is the absorbance of the sample. Sample concentration providing 50% inhibition (IC_50_) was obtained by plotting the inhibition percentage against sample concentrations.

### 4.5. Hydrogen Peroxide Scavenging Assay 

The hydrogen peroxide scavenging assay was adapted by Mukhopadhyay and coworkers [75] with some modifications. Briefly, 0.25 mL of ferrous ammonium sulfate (1 mM) solution was mixed with 1.5 mL of different concentrations (from 6 to 500 μg/mL) of carvacrol, *p*-cymene; acid gallic was used as a positive control. Then, 62.5 µL of H_2_O_2_ (5 mM) was added and the mixture was incubated at room temperature in the dark for 5 min. Then, 1.5 mL of 1,10-phenanthroline (1 mM) was added to each tube, mixed, and incubated for 10 min at room temperature. Finally, the absorbance was measured at 510 nm through a spectrophotometer (Thermo Scientific Multiskan GO, Monza, Italy).

### 4.6. Cell Cultures

Human neuroblastoma (SH-SY5Y) cancer cells (ATCC, Manassas, VA, USA (CRL-2266)) were cultured in Roswell Park Memorial Institute Medium (RPMI) supplemented with 1% l-glutamine, 10% heat-inactivated fetal bovine serum (FBS), and 1% penicillin/streptomycin at 37 °C in an atmosphere of 5% CO_2_.

### 4.7. Determination of the Maximum Non-Toxic Dose (MNTD)

The MTT assay determined the MNTD. SH-SY5Y cells (5 × 10^3^) were plated into 96-well culture plates (Corning, Inc., Corning, NY, USA) in 150 µL of culture medium and incubated at 37 °C in humidified 5% CO_2_. The cells were incubated for one day to allow for attachment and acclimatization. After 24 h, the cells were treated with different concentrations of carvacrol or *p*-cymene (5, 10, 25, 50, 100 μg/mL) for 24 h. Subsequently, 30 µL of 3-(4,5-dimethylthiazol-2-yl)-2,5-diphenyltetrazolium bromide (MTT) stock solution was added to each well, followed by three h of incubation at 37 °C in a dark environment to allow for the formation of purple formazan dye. After that, cells were lysed, and the dark blue crystals solubilized with 30 µL of a solution containing 50% *v*/*v N*,*N*-dimethylformamide and 20% *w*/*v* sodium dodecyl sulfate (SDS) and pH adjusted at 4.5. The optical density of each well was measured with a microplate spectrophotometer (Thermo Scientific Multiskan GO, Monza, Italy) equipped with a 520 nm filter. Cell viability in response to treatment was calculated as a percentage of control cells treated with DMSO at the final concentration of 0.1% [76]:% = (Abs treated well/Abs control well) × 100(2)

In order to determine the MNTD, a graph of the percentage of cell viability against the concentrations of substances was constructed.

### 4.8. Determination of the Optimal Concentration of H_2_O_2_

The growth inhibition of SH-SY5Y cells by H_2_O_2_ at concentrations of 100, 200, and 400 μM was determined using the method described in the previous paragraph; all concentrations of H_2_O_2_ were freshly prepared by diluting a H_2_O_2_ stock solution with RPMI. Cell viability in response to treatment was calculated as the percentage of control cells treated with DMSO at the final concentration (0.1%) after 2 and 4 h incubation with H_2_O_2_.

### 4.9. Determination of Neuroprotective Effects of Carvacrol and p-Cymene

To investigate the neuroprotective effects of carvacrol and *p*-cymene on H_2_O_2_-induced stress in SH-SY5Y cells, the cells were assigned to a total of nine groups: Group 1: control (untreated cells); Groups 2 and 3, cells treated with H_2_O_2_ alone, at 100 and 200 μM for two h; Groups 4–5: cells treated with carvacrol or *p*-cymene MNTD (50 µg/mL) Groups 6–9: cells pretreated with carvacrol or *p*-cymene MNTD for 22 h and then exposed to H_2_O_2_ at 100 and 200 μM for another two h. After the cell lysis, we determined the caspase-3 activity. For caspase-3 activity analysis, the SH-SY5Y cells (1 × 10^4^) were seeded in separate wells in 12-well plates (Corning, Inc., Corning, NY, USA). The cells were allowed to reach 70–80% confluency before the treatment groups were initiated. The cells were collected after 24 h and lysed using the Laemmli buffer to extract total proteins. For Western blot analysis, an aliquot of total protein was run on 10% SDS-PAGE gel and transferred to nitrocellulose. Nitrocellulose blots were blocked with 10% non-fat dry milk in Tris buffer saline 0.1% Tween-20 overnight at four °C and incubated with primary anti-caspase-3 antibody overnight and anti-GAPDH antibody for three h at room temperature. Immunoreactivity was detected through a sequential incubation with horseradish peroxidase-conjugated secondary antibody and enhanced chemiluminescence reagents [77]. Each band density was measured using ImageJ software (WS Rasband, Image J, NIH, Bethesda, MD, USA).

### 4.10. Statistical Analysis

All experiments were carried out in triplicate. Data from each experiment were statistically analyzed using GraphPad Prism 6.0 software (GraphPad Software Inc., San Diego, CA, USA), followed by a comparison of means (two-way ANOVA) using Dunnett’s multiple comparisons test at the significance level of *p* < 0.05.

## 5. Conclusions

The present study focused on the potential anti-acetylcholinesterase, anti-α amylase, and antioxidant activities of carvacrol and *p*-cymene, monoterpenes present in many essential oils from aromatic plants and widely studied for their numerous healthy properties. Although the recent literature showed common pathophysiological changes and signaling pathways between type-2 mellitus diabetes and Alzheimer disease, which were earlier considered as two independent disorders, no studies highlighted before the possibility of using natural substances as adjuvants in the treatment of both type-2 mellitus diabetes and Alzheimer disease. In particular, for those concerning anti-enzymatic activity, all IC_50_ values were less than 1 mg/mL, except for the activity of *p*-cymene against BChE. Instead, carvacrol, reducing caspase-3 expression, was clearly more neuroprotective than *p*-cymene. This last difference confirmed that the activity of carvacrol was due to the presence of an OH-group. Further studies are necessary in order to confirm the possible in vivo use of carvacrol or essential oils containing it as the main constituent as a coadjutant in preventing and treating AD in diabetic patients. 

## Figures and Tables

**Figure 1 ijms-24-06073-f001:**
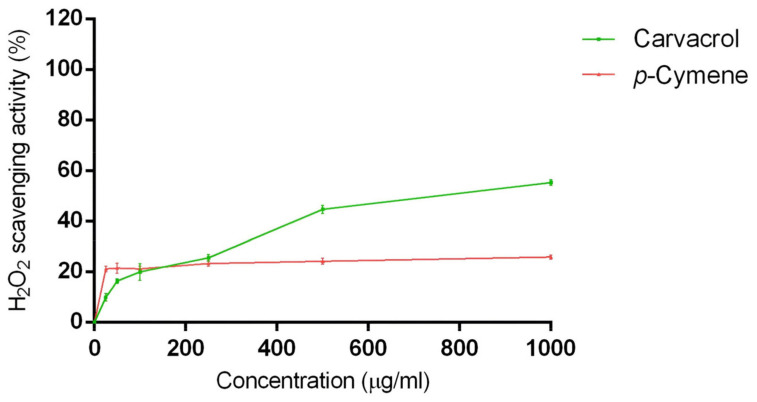
H_2_O_2_ scavenging action of carvacrol and *p*-cymene in the concentration range of 0–1000 μg/mL. Data are the mean ± SD of three experiments.

**Figure 2 ijms-24-06073-f002:**
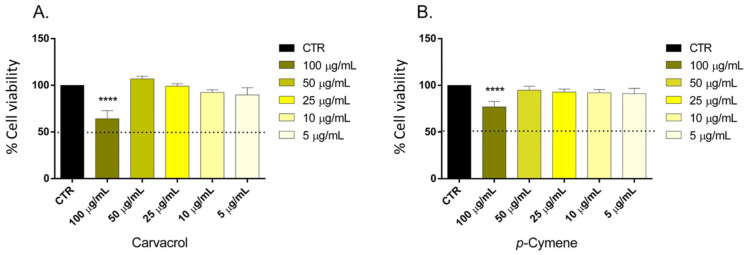
Cell viability calculated as a percentage after MTT assay. Cells were treated with different concentrations (5–100 μg/mL corresponding to 34–666 μM) of carvacrol (**A**) and *p*-cymene (**B**) for 24 h. Data are the mean ± SD of three experiments **** *p* < 0.0001 vs. control.

**Figure 3 ijms-24-06073-f003:**
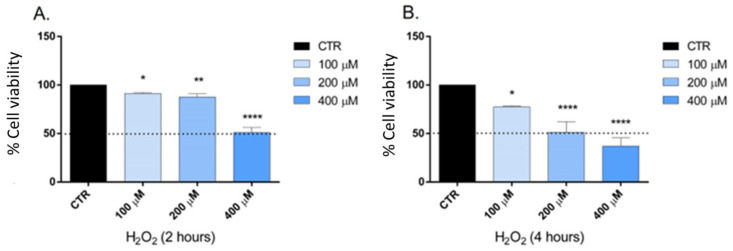
Cell viability calculated as a percentage after MTT assay. Cells were treated with different concentrations (100–400 μM) of H_2_O_2_ for 2 (**A**) and 4 h (**B**). Data are the mean ± SD of three experiments, * *p* < 0.05, ** *p* < 0.01, **** *p* < 0.0001 vs. control.

**Figure 4 ijms-24-06073-f004:**
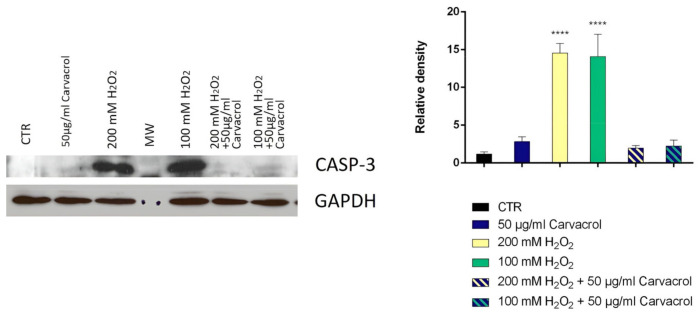
Relative expression levels of the caspase-3 protein in SH-SY5Y cells treated with carvacrol and H_2_O_2_. The panel shows the densitometry of bands in the treated group and control. Values are the mean ± SD in each group (n = 3). **** *p* < 0.0001 compared to control (ANOVA followed by Dunnett’s multiple comparison test).

**Figure 5 ijms-24-06073-f005:**
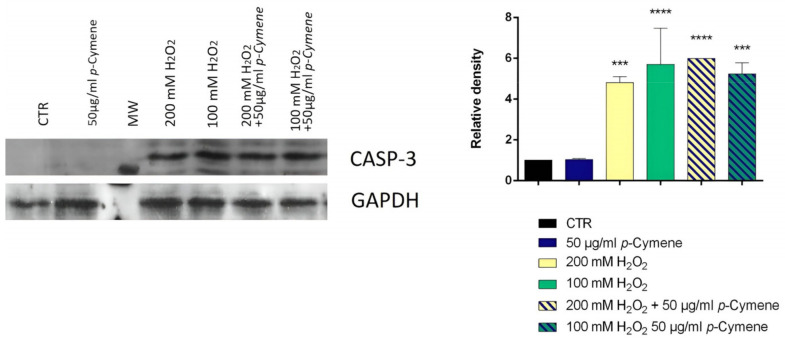
Relative expression levels of the caspase-3 protein in SH-SY5Y cells treated with carvacrol and H_2_O_2_. The panel shows the densitometry of bands in the treated group and control. Values are the mean ± SD in each group (n = 3). *** *p* < 0.001, **** *p* < 0.0001 compared to control (ANOVA followed by Dunnett’s multiple comparison test).

**Table 1 ijms-24-06073-t001:** Inhibitory effects of carvacrol and *p*-cymene on AChE, BChE, and α-amylase.

Compound	IC_50_ ^a^ (μg/mL)		
	AChE	BChE	α-Amylase
Carvacrol	3.8 ± 1.3 **^,####^	32.7 ± 5.5 ^####^	171.2 ± 10.8
*p*-Cymene	15.2 ± 3.6 ****	1456.0 ± 56.9 ****	215.2 ± 12.6
Galantamine	0.6 ± 0.3	4.5 ± 1.2	-
Acarbose	-	-	34.5 ± 6.4

^a^ IC_50_ = concentration required to reduce the enzymatic activity by 50%. Dunnett’s test vs. galantamine (** *p* < 0.01; **** *p* < 0.0001); vs. *p*-cymene (^####^ *p* < 0.0001).

## Data Availability

The data presented in this study are available on request from the corresponding author.

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
