# Peer review of "Anti-Cholinesterase and Anti-α-Amylase Activities and Neuroprotective Effects of Carvacrol and p-Cymene and Their Effects on Hydrogen Peroxide Induced Stress in SH-SY5Y Cells"

_ijms, 2023, doi:10.3390/ijms24076073_

Round 1

Reviewer 1 Report

The research conducted by Caputo and co-workers showed the possible anti-cholinesterase, anti-alfa-amylase, and neuroprotective effects of carvacrol and p-cymene. Their results suggest these compounds may act as neuroprotective against oxidative stress. 

The manuscript contains typographical errors, here are some:

Line 15: change respctively by respectively

Line 142: says it must say its

Line 170: The p in cimene should by italic

line 197: change th eneuroprotective by the neuroprotective

Line 250: N,N should by italic

Line 350: says proprieties must say properties 

I suggest reviewing the entire manuscript to correct it.

On the other hand, carvacrol is a well and extensively studied compound. For instance, the neuroprotection effect using neuroblastoma SH-SY5Y cells was described in 2015 by Bian et al. (Acta Pharmacologica Sinica), 2015, 36, 1426. Although the authors also studied p-cymene, there is a lack of novelty in this research. To improve the quality of the research, I strongly recommend supporting their findings with in vivo experiments. The manuscript and the results seem quite preliminary.

Besides, the authors say in line 89 "The AChE inhibitory effect exerted by carvacrol was 5 times stronger than that exerted by p-cymene, although carvacrol 89 and p-cymene have a very similar chemical structure. Furthermore, the IC50 of carvacrol 90 (3.8 μg/mL) is near to that of galantamine used as positive control (IC50 = 0.6 μg/mL).

There is an inaccuracy since carvacrol is 4 times stronger according to table 1. Then, the authors say that the IC50 value of carvacrol is near galantamine but the IC values are quite different, please be ethical when discussing the results. 

In section 3. Materials and Methods. The authors show that the purity of carvacrol is ≥ 97 and ≥90 for p-cymene. I wonder if it is necessary to improve the purity of p-cymene up to 97% since the compound is used to determine bioactivities.

I strongly suggest performing the in vivo experiments to confirm the possible use of carvacrol and p-cymene and being ethical when discussing the results to increase the quality of the research, without these experiments and a corrected discussion the manuscript should be rejected.

Author Response

Authors thank the Reviewer for its comments that improve the manusciript. Our responses are in the attached file. Please check.

Reviewer 2 Report

The aims of this study were: 1) to evaluate the possible effects of carvacrol and p-cymene on AChE, on BChE and α-amylase activities; 2) to study the potential of these monoterpenes as neuroprotective agents on hydrogen peroxide induced stress in SH SY5Y cells, considering that, in neurodegenerative diseases, a brain oxidative damage is often present.

The study is interesting. The authors need to describe in the aim some novelty of study.

Material and methods are described well.

Results and discussion must be written in separate chapters.

Conclusion is mainly described very generally.

Author Response

Authors thank the Reviewer for its comments that improve the manusciript

Reviewer 3 Report

Comments:

   In this manuscript, the authors described “Anti-cholinesterase, anti-a-amylase activities and neuroprotective effects of carvacrol and p-cymene and their effects on hydrogen peroxide induced stress in SH-SY5Y cells”. This paper show that Carvacrol and p-cymene showed in vitro anti-cholinesterase, anti-a-amylase activities and may act as neuroprotective agents against oxidative stress.  However, there are a few points that need to be clarified.

Comment

1.     Related monoterpene derivatives cymene, thymol and carvacrol have similar structures. Authors should provide relevant thymol data. Helps to understand the anticholinesterase, anti-α-amylase activity and neuroprotective effects of monoterpene derivatives.

2.     The introduction of related monoterpene derivatives is missing in the article.

3.     The article lacks the confirmation of animal experiments and the study of related mechanisms.

Author Response

(The authors gave the same response as above.)

Round 2

Reviewer 1 Report

The manuscript is well presented with an improved discussion/results/conclusions sections. All the comments/suggestions were attended, and although in vivo experiments will give robustness to the manuscript, it can be accepted as presented. Now is clear that the compounds studied may act as neuroprotective agents against oxidative stress. 

I suggest to accept the manuscript.

Reviewer 2 Report

All comments and suggestions were answered and changed.

Reviewer 3 Report

The author did not answer or make up relevant thymol data.